# Zein Multilayer Electrospun Nanofibers Contain Essential Oil: Release Kinetic, Functional Effectiveness, and Application to Fruit Preservation

**DOI:** 10.3390/foods13050700

**Published:** 2024-02-25

**Authors:** Farid Moradinezhad, Majid Aliabadi, Elham Ansarifar

**Affiliations:** 1Department of Horticultural Science, Faculty of Agriculture, University of Birjand, Birjand 9717434765, Iran; fmoradinezhad@birjand.ac.ir; 2Department of Chemical Engineering, Islamic Azad University, Birjand Branch, Birjand 9717711111, Iran; aliabadi@iaubir.ac.ir; 3Department of Nutrition & Food Hygiene, Social Determinants of Health Research Center, School of Health, Birjand University of Medical Science, Birjand P.O. Box 97175-379, Iran

**Keywords:** *Zataria multiflora*, electrospinning, multilayer fiber, zein, strawberry

## Abstract

In this study, sequential electrospinning was employed to produce a multilayer film consisting of zein nanofibers (Z) and Zataria multiflora essential oil (ZMEO) with different layers. The layers include: Z (without ZMEO), Z1 (one layer of Z + ZMEO), Z3 (three layers of Z + ZMEO), and Z5 (five layers of Z + ZMEO). Then, the effect of this antimicrobial packaging was investigated in relation to increasing the shelf life of strawberries at 4 °C for 12 days. The scanning electron microscopy (SEM) images of the fibers demonstrated a uniform and smooth structure without any beads. The use of Fourier transform infrared (FTIR) and Differential scanning calorimetry (DSC) showed that ZMEO was physically encapsulated into multilayer Z, resulting in an enhancement in thermal stability. The multilayer film showed a sustained release pattern of the encapsulated ZMEO for Z3, lasting for 90 h, and Z5, lasting for 180 h. This was in contrast to the rapid release within 50 h observed with Z film. The release kinetics for Z5 showed a good correlation with both the Higuchi and Korsmeyer–Peppas models, while for Z1 and Z3 films, Fickian diffusion was identified as the underlying mechanism. The findings of this study indicated that the multilayer film released ZMEO through a combination of diffusion and polymeric erosion. During a 12-day period of cold storage, strawberries that were treated with Z5 showed significant preservation of their anthocyanin (32.99%), antioxidant activity (25.04%), weight loss (24.46%), titratable acidity (11.47%), firmness (29.67%), and color (10.17%) compared to the control sample. The findings indicated that the sequential electrospinning technique used to create the multilayer nanofibrous film could be used in various fields, such as bioactive encapsulation, controlled release, antimicrobial packaging, and food preservation.

## 1. Introduction

The strawberry (*Fragaria ananassa*) is a popular non-climacteric fruit that is recognized for its nutritional value and sensory properties. Strawberries are also very delicate and prone to spoilage because they can easily be damaged and they suffer from post-harvest physiological problems and fungal infections [1]. Due to their short shelf life, these fruits are considered a fragile commodity, and it is estimated that up to 40% of them are lost during post-harvest storage. The shelf life of a food item refers to the duration during which it remains safe to consume and retains its intended sensory, chemical, physical, microbiological, and functional properties without any changes, as specified on the label, when stored under the recommended conditions [2]. Fruits are nutritious and contain fibers, minerals, and vitamins. They are made up of living tissues and have a high water content, which makes them prone to microbial contamination and quick spoilage [3]. Extending the shelf life and preserving their properties are challenges for fruit packaging systems [4]. Antimicrobial packaging is a successful technique for increasing the shelf life of food items while maintaining their nutritional and sensory properties [1,5]. It is essential in preventing the growth of certain microorganisms on food, which improves food safety and extends shelf life without affecting the quality of the food [6].

Essential oils have antibacterial, antifungal, and antioxidant properties. The Food and Drug Administration (FDA) considers them safe, and they can be used as natural food additives and prevent the growth of harmful bacteria [6]. Zataria multiflora Boiss., also known as Avishan-e-Shirazi, is a plant similar to thyme that is native to Iran, Pakistan, and Afghanistan. It is highly regarded for its medicinal and culinary properties [1,5]. The food industry has increasingly adopted the utilization of natural compounds, such as essential oils, for their beneficial properties, including antioxidant activity and antimicrobial effects, as well as their appealing taste, as preservatives and food additives [7,8]. Essential oils are highly susceptible to environmental factors that can cause their deterioration and affect their taste [6]. Encapsulation is a useful technique that can help to protect essential oils from degradation, while also masking any unpleasant flavor. This approach can significantly improve the longevity and effectiveness of essential oils [8]. 

Electrospinning has gained popularity as a convenient and effective method for encapsulating essential oils [9]. Electrospun fibers are popular for creating films and active packaging compared to traditional casting or coating methods. Nanoparticles can release encapsulated contents in a controlled manner due to their high surface-area-to-volume ratio, small pores, and increased porosity [1,4,8].

The objective of this research was to produce a multilayer nanofiber using the electrospinning technique with varying numbers of layers and to analyze the release kinetics of Zataria multiflora essential oil. This study also aimed to assess the potential of these nanofibers in prolonging the shelf life and maintaining the quality of strawberries.

## 2. Materials and Methods

### 2.1. Materials

In this study, the materials utilized were zein, which was obtained from Sigma, (Saint Louis, MA, USA) (CAS Number 9010-66-6), Zataria multiflora essential oil (ZMEO) from Barij Co. (Tehran, Iran), and glacial acetic acid obtained from Sigma (CAS Number 64-19-7). A local farm in Birjand, Iran was the source of the organic strawberries that were bought.

### 2.2. Gas Chromatography–Mass Spectrometry (GC–MS)

The ZMEO was analyzed using GC–MS. A DB5 capillary column (30 mm, 0.25 mm, 0.25 μm) was used in the chromatograph. The information was obtained with the given parameters: starting temperature of 50 °C, ending temperature of 250 °C, ionization energy of 70 eV, and the use of helium as a carrier gas with a flow rate of 1.1 mL/min [10]. 

### 2.3. Solution Preparation

The zein solution with a concentration of 30% (*w*/*v*) was prepared by dissolving zein in glacial acetic acid at a temperature of 25 °C. The solution was then stirred for a period of 24 h until a uniform and consistent solution was achieved. The solution was mixed with Zataria multiflora essential oil (ZMEO) at a concentration of 4% *w*/*v* for a duration of 24 h.

### 2.4. Electrospinning

Nanofibers were produced using electrospinning with a high-voltage machine (model ES-1000, Nanoscale Technologists CO., Tehran, Iran). The capillary needle was connected to the positive and negative electrodes for this purpose, while the rotating collector was wrapped with aluminum foil. The machine parameters were set as follows: voltage at 15 kV, rate of injection at 0.5 mL/h, and the distance between the needle tip and the collector at 15 cm.

### 2.5. Characterization of Nanofibers

#### 2.5.1. Scanning Electron Microscopy (SEM)

The scanning electron microscope (Leo 1450VPSEM, Ted Pella, Redding, CA, USA) was used to observe the morphology of the multilayer fibers, with an acceleration voltage of 20 kV.

#### 2.5.2. Fourier Transform Infrared (FTIR)

The chemical interactions of functional groups in multilayer fiber were evaluated by conducting FTIR spectra. The samples were analyzed using a Fourier transform spectrophotometer (Shimadzu 6650, Kyoto, Japan) with a resolution of 4 cm^−1^ in the frequency range of 400 to 4000 cm^−1^.

#### 2.5.3. Differential Scanning Calorimetry (DSC)

Differential scanning calorimetry (NETZSCH STA 449F3, Selb, Germany) was used to assess the thermal stabilities of multilayer fibers. A 5 mg sample was subjected to heating from 20 to 300 °C at a rate of 10 °C/min while being exposed to a continuous flow of nitrogen gas at 20 mL/min.

#### 2.5.4. Encapsulation Efficiency and Release of ZMEO from Zein Multilayer Nanofiber

The entrapment efficiency (EE) of the ZMEO-loaded nanofiber was indirectly determined to dissolve the film (5 g) in 1 mL ethanol aqueous solution (50%) with a shaking speed of 120 rpm for 10 h. The entrapment efficiency (EE) of ZMEO loaded into nanofiber was determined indirectly using the following method: the film was dissolved in 1 mL of a 50% ethanol aqueous solution and shaken at a speed of 120 rpm for 10 h. The concentration of ZMEO in the film was determined by the absorbance at 278 nm. The equation of the EE was calculated as follows:(1)EE%=WiWt×100
where *W_i_* is the actual ZMEO amount and *W_t_* is the theoretical ZMEO amount.

The zein/ZMEO nanofiber film was kept in a closed container under normal room temperature, and the quantity of remaining ZMEO was measured after every 2 intervals. The amount of ZMEO that was released from the nanofiber film was assessed by dissolving 20 mg of the film in 20 mL of 50% ethanol and determining the remaining amount of ZMEO using UV spectroscopy at 278 nm for a duration of 180 h [11].

The models of Higuchi and Korsmeyer–Peppas were described by Equations (2) and (3), respectively.
(2)MtM∞=kt1/2
(3)MtM∞=ktn
where *M_t_* and *M*_∞_ represent the mass of ZMEO released at time *t* and the mass of ZMEO at equilibrium, respectively; *k* is the release rate constant; and *n* is the release exponent.

### 2.6. Fruit Preparation and Active Packaging

The laboratory selected strawberries based on uniformity in size, ripeness, color similarity, appearance, and the absence of surface defects and fungal spoilage after the fruits were transferred. The strawberries were treated by immersing them in a 0.1% sodium hypochlorite solution for 1 min, followed by dripping off for 5 min. The strawberries were divided into five groups, with seven strawberries per group for each replicate. Finally, they were packed in polyethylene containers with dimensions of 11.5 × 9.5 × 6.2 cm and sealed with their lids. The experiment involved packing a control group of fruit in PET containers without a lid film, while other treatment groups used PET containers with Z/ZMEO, Z1/ZMEO, Z3/ZMEO, and Z5/ZMEO nanofiber films on the lids. All fruit packages were stored in a cold room at 4 ± 0.5 °C and 85% RH for 12 days. The physicochemical properties of the fruits were measured at different intervals (day 0, 3, 6, 9, and 12).

### 2.7. Evaluation of Strawberry Quality

Weight loss: The digital balance (UWA-K-015, Xiamen Andong Electronics Co., Ltd., Xiamen, China) was used to weigh the fruits, and the outcome was determined by calculating the percentage of weight lost between the initial and final measurement [12].

Total soluble solids (TSS): The fruits were crushed using a pestle, and the resulting juice was collected. The juice’s total soluble solids (TSSs) were measured using a hand-held refractometer (Atago, Tokyo, Japan) at a temperature of 20 °C.

Titratable acidity (TA): The juice’s titratable acidity (TA) was determined by adding 0.1 N of NaOH to the diluted juice (5 to 95) until the pH reached 8.2 [12]. 

Total anthocyanin content: The measurement of anthocyanin contents in strawberries was performed using the pH differential method described by Wang et al., (2022) [13]. The strawberries were mixed well with 20 mL of pH 1.0 buffer (0.025 M potassium chloride) and pH 4.5 buffer (0.4 M sodium acetate) and incubated for 20 min at room temperature. 

Antioxidant activity: The absorbance was measured at 520 and 700 nm. The 2,2-diphenyl-1-picrylhydrazyl (DPPH) radical scavenging assay is a common method used to measure the antioxidant activity of fruits. In the assay, a solution of DPPH is mixed with a sample of the fruit extract being tested. The DPPH radical reacts with the antioxidants in the fruit extract, causing a decrease in the absorbance at 517 nm. The degree of decolorization is proportional to the antioxidant activity of the fruit extract [6].

Firmness: The firmness of the strawberries was measured utilizing a digital penetrometer (FHT200, Extech Co., Mansfield, TX, USA) equipped with a cylinder probe with a 2 mm diameter. The results were expressed in Newton.

Color parameter: The Colorimeter (TES 135-A, Taipei, Taiwan) was used to measure the surface color attributes of the strawberries. The values of L* (light/dark), a* (red/green), and b* (yellow/blue) were recorded from both sides of each [12].

### 2.8. Statistical Analysis

The distinction between samples was assessed using analysis of variance (One-way ANOVA with Duncan test) at a significance level of *p* < 0.05. A post hoc test for independent samples was conducted using the statistical software SPSS (IBM SPSS Statistics, Version 22, New York, NY, USA).

## 3. Results and Discussion

### 3.1. Essential Oil Analysis by GC–MS

Table 1 displays the chemical composition of the ZMEO. In total, among the 15 compounds identified through GC–MS, the essential oil primarily consisted of thymol, accounting for 43.84%, and this was followed by terpinene (22.46%), p-cymene (14.32%), and carvacrol (10.94%). ZMEO contains high levels of thymol and carvacrol, which possess effective antibacterial and antioxidant properties [10]. The differences observed could be attributed to the geographic location or the ecotype of the plants that were gathered. Carvacrol, one of the phenolic compounds found in ZMEO, exhibits the strongest antimicrobial activity. This is attributed to its hydrophobic properties and the presence of a free hydroxyl group, which is crucial for its effectiveness in targeting cell membranes [10].

### 3.2. SEM

The SEM images of multilayer zein fibers with and without the ZMEO are shown in Figure 1. As expected, with the increasing number of layers, the average fiber diameters significantly increased (*p* ˂ 0.05), and the mean fiber diameters of the Z, Z1, Z3, and Z5 samples were 243 ± 31 nm, 387 ± 32 nm, 432 ± 52 nm, and 548 ± 35 nm, respectively. The alteration can be linked to a rise in the electrostatic repulsion among charges present on the surface of the jet while electrospinning the zein solution. As a result, this led to the creation of fibers that were marginally thinner. Also, as displayed in Figure 1, all of the electrospun fibers had a linear and homogeneous morphology, a smooth surface, and a bead-free structure. Multilayer fibers (especially Z5) exhibited an increase in fiber network density with a nearly ribbon-like shape. Shao et al., (2019) [8] reported that the zein interlayer displayed a strong connection to the outer layers in the multilayer structure due to the hydrophobic interactions.

The addition of ZMEO to fibers resulted in a significant increase in their diameter (*p* < 0.05) while still maintaining their smooth and bead-free morphology without any collection. This confirmed that the encapsulation of ZMEO in the zein did not affect fiber formation. Karim et al., (2020) [6] and Vafania et al., (2019) [14] reported that increasing essential oil in a polymer solution (thyme in chitosan–gelatin and cinnamic aldehyde in zein, respectively) led to an increase in the diameters of the nanofibers. This is probably attributable to the decrease in electrical conductivity, which reduces the elongation of the polymer jet through the applied voltage [6,14].

### 3.3. FTIR

The FTIR spectra of zein powder, ZMEO, and multilayer nanofibers Z, Z1, Z3, and Z5 between 400 and 4000 cm^−1^ are presented in Figure 2. The FTIR was applied to study all of the interactions happening between the ZEO, zein, and multilayer nanofibers. The characteristic peaks of zein powder were observed at 1535 cm^−1^ (amide II), 1451 cm^−1^ (C-N stretching), 2872 and 2959 cm^−1^ (carboxylic acids group), 1658 cm^−1^ (C=O stretching vibration of amide I), and 3323 cm^−1^ (−OH group) (Figure 2) [8,9]. The characteristic peaks of ZMEO were observed at 2961 cm^−1^ and 2871 cm^−1^ (asymmetric and symmetric methyl C–H stretching), 1289 cm^−1^ (aromatic ether), 1619 cm^−1^ (conjugated double bond of the ring), 3525 cm^−1^ (O–H vibration of hydroxyl group of thymol), 1516 (C=C aromatic ring), 810 cm^−1^ (CH wagging vibrations), and 1584 cm^−1^ (N–H bending) [7,14] (Figure 2). The zein nanofiber without ZMEO displayed all of the characteristic peaks of zein. This indicates that there were no chemical changes in the polymer resulting from electrospinning. The spectra of Z1 (zein nanofiber with ZMEO), Z3 (zein nanofiber with two layers), and Z5 (zein nanofiber with five layers) displayed all of the characteristic peaks of the zein and ZMEO. The reason for this is that the incorporation of ZMEO into zein nanofiber was a purely physical process, with no chemical interaction taking place between zein and ZMEO. According to Vafania et al., (2019) [14], Karim et al., (2020) [6], and Lin et al., (2018) [7], the use of chitosan–gelatin and zein nanofibers for encapsulating essential oils did not result in any significant alteration of the polymer or the oil’s characteristic peaks. This confirms that successful encapsulation was achieved through physical interactions.

### 3.4. DSC

The DSC thermograms of powder zein, ZMEO, and multilayer nanofiber (Z, Z1, Z3, and Z5) are shown in Figure 3. The ZMEO exhibited a single endothermic peak at a temperature of 196 °C, which corresponds to its melting point [1]. The zein powder exhibited a wide endothermic peak centered at approximately 95.5 °C and another broad endothermic peak at 296 °C, indicating the complete unfolding of the zein structure [15]. The nanofiber Z exhibited a wide endothermic peak, reaching its maximum at approximately 100 °C. This peak was caused by the release of water trapped within the polymer and the disruption of hydrogen bonds. Additionally, there was a broad endothermic peak observed at 310 °C, indicating the unfolding of the protein structure [8,15]. The starting point of main weight loss of multilayer nanofibers (Z1, Z3, and Z5) was more than pure ZMEO, which provided a thermal stability improvement. The encapsulation of ZMEO into zein nanofiber has been shown to delay thermal degradation, which is attributed to the natural antioxidant properties of the essential oil [8,16]. Lin et al., (2018) [7] also found a comparable outcome when they created a gelatin-based active film with thyme essential oil. They noted that the gelatin films containing thyme exhibited greater thermal stability compared to the pure gelatin film. The observed enhancement in thermal stability can be attributed to the hydrogen bonding interactions between the polymer chain’s terminal −OH groups and the −OH, −CO, and −COOH chemical groups of the polymer components [16]. It is worth noting that the thermal stability of the Z3 and Z5 multilayer nanofibers was marginally superior to that of the Z1 monolayer, indicating that the number of layers of zein nanofibers was more effective for the thermal stability of ZMEO.

### 3.5. Encapsulation Efficiency (EE), Release Analysis, and Kinetic Modeling

The EEs of ZMEO in the zein multilayer nanofibers (Z1, Z3, and Z5) were 43%, 56%, and 82%, respectively. It was clear that as the number of outer layers increases, the EE significantly increases. According to Zhang et al., (2020) [4], when using optimal electrospinning conditions, the biodegradable nanofiber (poly lactide-co-glycolide) achieved a high loading capacity (LC) of 8.61% and an encapsulation efficiency (EE) of 90.3% for thymol [4].

The release profiles of ZMEO from the multilayer films Z1, Z3, and Z5 are displayed in Figure 4. It can be seen that ZMEO in the Z1 nanofiber film released quickly at room temperature, and 78% of it was released after 50 h. The release of ZMEO encapsulated in the Z3 nanofiber film was slower, and 72% of the ZMEO was released into the atmosphere after 90 h, while 68% of the ZMEO encapsulated into the Z5 nanofiber film was released after 180 h. This was probably because zein multilayer nanofibers formed a core–shell structure through the process of electrospinning and effectively controlled the release of ZMEO [11]. Zhang et al., (2020) [4] reported that the control group showed rapid evaporation, with complete evaporation of thymol occurring after 19 h. In contrast, the release of thymol from the core–shell nanofiber film was significantly slower, with only 36% of the encapsulated thymol being released into the atmosphere after 72 h [4]. Two commonly used models (namely, Higuchi and Koremeyer–Peppas) were used to assess the release mechanism of ZMEO from zein multilayer films (Z1, Z3, and Z5). Table 2 presents the kinetic parameters, while R2 and RMSE values were used to assess the effectiveness of the models, with higher R2 and lower RMSE values indicating better fitness [11]. The release profile of ZMEO from Z1 and Z3 films was fitted using the Korsmeyer–Peppas (R^2^ = 0.963, RMSE = 0.186) (R^2^ = 0.966, RMSE = 0.234) model, and the kinetic parameters were 0.065 and 0.83 (n) and 4.739 and 2.325 (k), respectively. This indicated that the release of ZMEO from Z1 and Z3 films was the mechanism of Fickian diffusion. The release kinetics for the multilayer (Z5) exhibited a good correlation with both the Higuchi (R^2^ = 0.981, RMSE = 0.241) and Korsmeyer–Peppas (R^2^ = 0.963, RMSE = 0.321) models. The findings indicated that the release of ZMEO from the multilayer film was linked to a combination of diffusion and polymeric erosion. This was supported by the Korsmeyer–Peppas model, which explained the mechanism of non-Fickian diffusion, and the Higuchi model, which demonstrated the release of ZMEO through a porous matrix [11].

### 3.6. The Effect of Active Packaging on Weight Loss Percentage

As shown in Table 3, active packaging treatments affected strawberry fruit weight loss during storage (4 °C and 80 ± 5% RH). The use of all active packaging treatments compared to the control caused the weight loss to be controlled, such that the highest amount of weight loss was obtained from the control (8.34%) and the lowest amount of weight loss was related to the Z5 + ZMEO (6.30%). With increasing storage time, the amount of weight loss in strawberry fruits increased. However, the interaction effects of storage time and packaging treatments presented in Figure 5 showed that the active packaging treatments had less weight loss than the control. It was also shown that by adding ZMEO to the coating and increasing the number of coating layers, the amount of weight loss of strawberry fruit was significantly reduced. As such, the Z5 + ZMEO treatment during the storage period reduced the weight loss of strawberries by 20–30% compared to the control.

Weight loss in fruits is primarily linked to the respiration and evaporation that occur through their skin, making the characteristics of the fruit skin significant [2]. The thin skin of strawberry fruits makes them prone to rapid water loss, causing them to shrivel and spoil. The amount of water that is lost is determined by the difference in water pressure between the fruit tissue and the air around it, as well as the temperature at which the fruit is stored [3]. Therefore, storage in conditions of high humidity (80 to 90%) and low temperature is recommended for strawberry storage [17]. Dehydration also increases surface-scarred fruits. Damaged fruits are susceptible to contamination and spoilage. Proper packaging limits the transfer of water from the skin of the fruit to the surrounding space [18]. Preventing moisture loss and reducing weight loss protect fruits against mechanical damage. The result of this process is probably having fresh, juicy, and free-of-microbial-contamination fruit at the end of the storage period. As mentioned, the increase in respiration rate increases the weight loss of the fruit [17]. One of the reasons for fruit weight loss during storage in cold storage is contamination with pathogenic agents, which, by damaging the fruit tissue, increases the rate of respiration and increases the weight loss rate of fruits [17]. Therefore, ZMEO present in the zein fibers prevents the weight loss of the fruit by reducing post-harvest pollution due to its antimicrobial properties. According to the results obtained from this research, Moradinezhad et al., (2020) [18] reported that the thymol released from the fibers slowed down the ripening process more effectively and, as a result, significantly reduced the weight loss of strawberry fruit. Also, Perdones et al., (2012) [17] showed that the use of essential oils in the packaging of strawberry fruits causes a decrease in oxygen and an increase in carbon dioxide in the package containing the fruit. They explained that the essential oil probably affects the metabolic pattern of strawberry and modifies the respiratory behavior, so the weight loss characteristic of the fruit is affected by the respiration rate of the fruit.

### 3.7. The Effect of Active Packaging on Firmness (N)

The quality of stored fruit is determined by firmness, and a decline in firmness is an indicator of a decline in fruit quality. A successful packaging must maintain the firmness of the fruit. Table 3 shows the firmness values of strawberry fruit tissue. As expected, the firmness of the strawberry fruit texture decreased with an increasing storage period. It was shown that on the day of harvest, the firmness of the fruit tissue was 1.90 N. Meanwhile, at the end of the storage period (day 12), this value reached 0.95 N, with a decrease of about 50%. Although the amount of fruit tissue firmness decreased during the storage period, the investigation of active packaging showed that the Z5 + ZMEO treatment maintained more fruit strength than the control. Also, investigating the mutual effects of storage time and active packaging showed that all active packaging treatments had more tissue firmness than the control on the days of measurement (3, 6, 9, and 12). The results showed that the control group experienced a significant decrease in firmness compared to the other experimental groups on the 3rd day of the experiment. This strong decrease in firmness or increase in texture softness in strawberry fruit can be attributed to the increased respiration of the control group compared to the other treatments (Figure 5).

During ripening, the cell walls of parenchyma undergo significant deformation, resulting in changes to their mechanical properties and a considerable reduction in cell adhesion due to the breakdown of the middle lamella [19]. Usually, the firmness of strawberry fruit tissue during the storage period is reduced due to the dissolution and destruction of pectin compounds and other cell wall compounds through the activity of certain enzymes, such as polygalacturonase and pectin methylesterase [2]. Meanwhile, essential oils maintain the firmness of the fruit tissue by reducing the activity of cell wall enzymes and, as a result, reducing the production of soluble pectins and converting insoluble pectins to soluble ones [20]. It is possible that ZMEO enclosed in fibers slows down the catabolic processes due to ripening and ultimately preserves the firmness of the fruit tissue [21]. Also, as it can be seen, as the number of layers increases, there is a significant decrease in the softening of strawberries. Although there are many reports of the effect of essential oils on fruits, their mechanism of action is not fully known. Valero et al., (2013) [20] discovered that by using carvacrol, eugenol, and thymol essential oils for packaging, the firmness of plum fruit tissue can be maintained at about 20% higher than the control. They suggested that the essential oils likely impact the ethylene intermediate compounds and delay the ripening of the fruit, thus preserving the firmness of the texture. Rahimi et al., (2019) [21] reported a similar account of the use of thymol on peach fruit.

### 3.8. The Effect of Active Packaging on Total Soluble Solids (TSSs)

Examining the changes of total soluble solids (TSS) during strawberry storage showed that active packaging had no effect on TSSs. However, different storage times (0, 3, 6, 9, and 12 days) had a significant effect on the TSS of strawberry fruit. According to the data reported in Table 3, with the passage of time (until the 6th day of storage), the level of TSS increased, and after 6 days until the end of the storage period (12 days), the TSS of strawberry decreased. The decrease in the amount of TSS of strawberry compared to the first day of the experiment (0 day) was about 14%. As shown in Figure 5, examination of the interaction effects of storage time and active packaging revealed a decrease in the TSS amount during storage. However, this reduction was less observed in the treatments that were packaged with ZMEO enclosed in Z1, Z3, and Z5 fibers. The results show that the use of active packaging (combined with ZMEOl) preserved the soluble solids in strawberry fruit.

The results of the present research are consistent with the results of Geransayeh et al., (2015) [22]. The reduction of TSS in strawberry fruit can be related to high fruit metabolism and senescence processes [18,22]. The lower rate of respiration in packed strawberries is probably due to the presence of ZMEO, which helped to maintain higher levels of carbohydrates in the tissue [20]. Also, Perdones et al., (2012) [17] stated that the level of TSS in strawberries increased due to water loss during storage, and the fruits that experienced the most water loss showed greater changes in TSS, which aligns with our findings.

### 3.9. The Effect of Active Packaging on Titratable Acidity (TA)

The TA levels of strawberry fruit in different active packages are reported in Table 3. The numerical range of strawberry TA in this research was reported as 0.54 to 0.61%. The results showed that the use of ZMEO enclosed in Z5 significantly maintained the TA of strawberry fruit compared to the control and other treatments. According to the data reported in Table 3, the amount of TA in the Z5 + ZMEO treatment was 12% higher than the control. Also, investigations showed that the amount of TA in strawberries decreased during storage. On the first day of the test, the value of TA was 0.74%, while at the end of storage (day 12), this value decreased by 35% to 0.48%. The results of interaction effects of storage time and active packaging are shown in Figure 5. The investigation of mutual effects showed that although the amount of TA of strawberry fruit decreased during the storage time, the use of active packaging delayed the reduction of TA. 

The most prevalent acid found in strawberries is citric acid. In general, acids play a role in the regulation of cell pH, and it is probable that the stability of anthocyanin and, as a result, the color of the fruit are related to acids [13]. It is thought that the consumption of organic acids and fruit aging are effective factors in reducing acidity during storage [13]. Various treatments, such as the type of packaging that leads to reduced respiration, can delay aging and thus the rate of use of organic acids [23]. It is also important to note that the use of essential oils in packaging, which often have antimicrobial properties, reduces the rate of fruit deterioration due to the prevention of the spread of bacterial and fungal contamination; finally, they are preserved by controlling the respiration of organic acids [20]. Some researchers have associated the decrease in TA with the loss of water in the fruit [22].

### 3.10. The Effect of Active Packaging on the Total Anthocyanin Content

The treatments applied during the storage period had a significant impact on the overall amount of anthocyanin present in strawberries (Table 4). Strawberries stored in active packages showed the highest anthocyanin values compared to the control. However, the fruits of Z5 + ZMEO treatment had the maximum anthocyanin content. Interestingly, the results showed that the use of ZMEO enclosed with Z5 fibers during the post-harvest period maintained the anthocyanin content about 50% more than the control fruits. Also, by increasing the layers of zein fibers combined with ZMEO in packaging, the anthocyanin content increased. Over time, the total anthocyanin content in strawberry fruits decreased. However, the results of interaction effects of storage time and active packaging showed that active packaging reduced the degradation of anthocyanins during the storage period (Figure 6).

Anthocyanins are produced as secondary metabolites primarily via the shikimate, phenylpropanoid, and flavonoid pathways, as well as the pentose phosphate pathway [13]. Anthocyanin synthesis involves multiple enzymatic steps, with each step being catalyzed by a sequential reaction. Among the key enzymes in the destruction of anthocyanins are polyphenol oxidase and peroxidase enzymes, which probably led to the reduction of these enzymes and the increase in the accumulation of anthocyanin content. The results obtained from the present study were consistent with the results of other researchers [20].

### 3.11. The Effect of Active Packaging on the Antioxidant Activity

The antioxidant activity of strawberry fruit depends on pre-harvest and post-harvest conditions, such as environmental factors, harvesting time, and storage conditions. As shown in Table 4, active packaging (Z5 + ZMEO) increases the antioxidant activity of strawberry fruit by about 34% compared to the control. Investigating the effect of storage time on the antioxidant activity showed that the antioxidant level increased strongly until the 6th day of the experiment, and then the antioxidant activity decreased until the last day of storage (12th day). However, the fruits at the end of storage had about 93% more antioxidant activity than on the first day of the experiment. According to Figure 6, an increase in antioxidant activity was observed in all treatments compared to the first day of the experiment. However, it was clearly shown that the fruits wrapped with five-layer zein fiber combined with thymol was the most effective treatment to preserve the antioxidant content of strawberry fruit for 12 days. Also, the control showed the lowest antioxidant retention (Figure 6).

The antioxidant activity of products is reduced during the post-harvest stage mainly due to oxidation. Antioxidants help eliminate free radicals that are formed during oxidation, leading to the loss of antioxidants over time during storage [23]. The activity of oxidative enzymes, such as peroxidase or ascorbate oxidase, depends on the concentration of O2 [24]. In this study, the use of ZMEO enclosed with zein fibers caused the accumulation of antioxidant activity. In this case, the release of ZMEO from the zein fibers during storage can help reduce contamination and deterioration of the fruit. As a result, the respiration of the fruit decreases, ultimately reducing the amount of oxygen consumption. This is an important factor in decreasing oxidative damage [24]. With the reduction of oxidative activity, the use of antioxidants to neutralize free radicals decreases, so the content of antioxidants accumulates and increases in the tissue. The results of this research are consistent with the results of other studies [1].

### 3.12. The Effect of Active Packaging on Color Parameter (L*, a*, and b*)

The color of fruits is a crucial characteristic, and it plays a significant role in determining the quality of the fruit and its value in the market. The surface color of strawberry fruit was investigated. According to the results presented in Table 4, the highest amount of *L**, *a**, and *b** was obtained from the Z5 + ZMEO treatment, followed by the Z3 + ZMEO treatment. However, all three color parameters (*L**, *a**, and *b**) decreased during 12 days of storage. *L** values varied from 47.21 on the first day of the experiment to 34.65 at the end of storage. 

Fresh products lose their moisture over time due to breathing and aging during storage, which reduces their color quality [12]. However, we found that the use of active packaging preserves the color quality of the fruit during storage (Figure 7). The reduction of *L** indicates the dull color of the fruit skin and the reduction of the luster and freshness of the fruits [25]. The present study showed that although the amount of component *L** in the color of strawberry skin decreases with time, the fibers used in packaging (zein and ZMEO) lead to maintaining the brightness and freshness of the fruits.

The levels of *a** in the color of the fruit indicate the degree of redness [26]. Anthocyanin pigments (specifically, pelargonidin derivatives) are responsible for the vibrant red color of strawberries [13]. Color deterioration is accompanied by loss of anthocyanins, and this detrimental change persists during storage of strawberries. The fluctuation in color of strawberry products can be caused by various factors, such as enzymatic browning, Maillard browning, degradation of ascorbic acid catalyst, and polymerization reactions of anthocyanins with other substances [27]. For a while now, it has been understood that the degradation of anthocyanins is influenced significantly by ascorbic acid [26,27]. The oxidation of ascorbic acid leads to the production of free radicals that can break the pyrylium ring of anthocyanins. It has also been hypothesized that ascorbic acid can condense with anthocyanins at the four-position, leading to the loss of flavylium pigmentation [27]. Degradation of anthocyanins by ascorbic acid can occur in both aerobic and anaerobic conditions. Anthocyanins can be degraded indirectly by polyphenol oxidase through the oxidation of phenols to quinones, which subsequently combine with anthocyanins to produce dark brown compounds [3]. Following this reaction, the levels of *a** (red) decrease drastically. Probably, in this research, ZMEO enclosed in zein layers reduces free radicals and prevents the destruction of anthocyanin and the reduction of the level of *a** in strawberries. Our findings were consistent with the results of other researchers [1,27]. 

## 4. Conclusions

Passive and active packaging are undoubtedly some of the most accessible and promising industrial applications associated with electrospinning technology. This study showcases the use of sequential electrospinning to create a multilayer film composed of zein nanofibers (Z) and Zataria multiflora essential oil (ZMEO) in various layers. The film had different layers, including Z (without ZMEO), Z1 (one layer of Z + ZMEO), Z3 (three layers of Z + ZMEO), and Z5 (five layers of Z + ZMEO). The film was then tested for its effect on increasing the shelf life of strawberries at 4 °C for 12 days. The film showed a sustained release pattern of the encapsulated ZMEO for Z3, lasting for 90 h, and Z5, lasting for 180 h, compared to the rapid release within 50 h observed with Z film. This study suggests that the multilayer film releases ZMEO through a combination of diffusion and polymeric erosion. Z5-treated strawberries showed significant preservation of their anthocyanin, antioxidant activity, weight loss, titratable acidity, firmness, and color during a 12-day cold storage period compared to the control sample. This technique has the potential to significantly contribute to the development of innovative multifunctional materials for food packaging. The results suggest that the multilayer nanofiber film created using the sequential electrospinning technique has potential applications in different areas, like bioactive encapsulation, controlled release, antimicrobial packaging, and food preservation.

## Figures and Tables

**Figure 1 foods-13-00700-f001:**
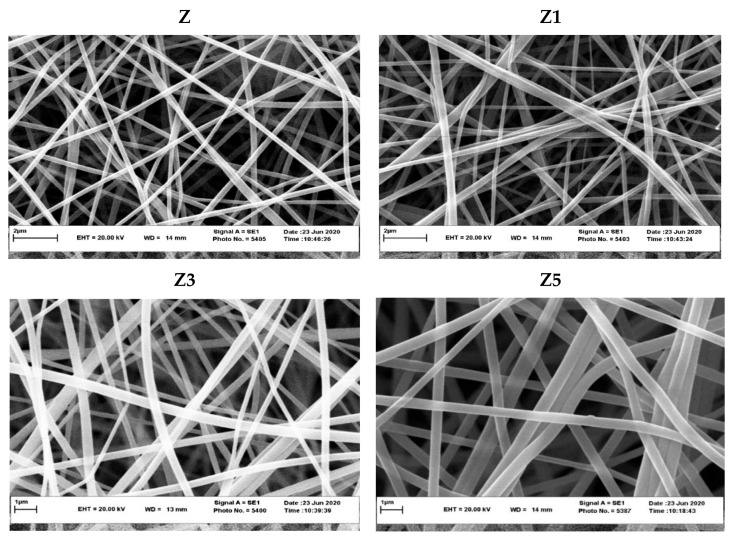
SEM images of Z (zein fiber without ZMEO), Z1(one-layer zein fiber with ZMEO), Z3 (three-layer zein fiber with ZMEO), and Z5 (five-layer zein fiber with ZMEO) (mean ± SD; means with different letters are statistically significant; *p* ˂ 0.05).

**Figure 2 foods-13-00700-f002:**
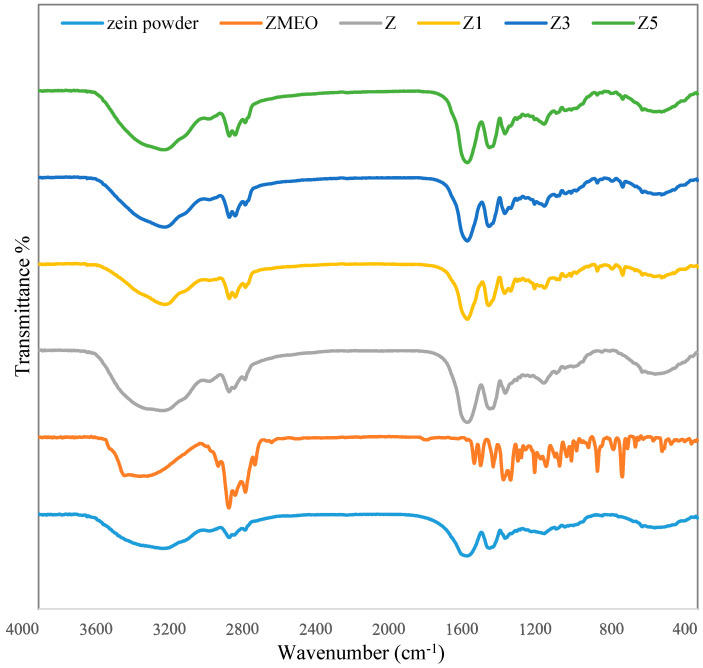
Fourier transform infrared (FTIR) spectra of Z (zein fiber without ZMEO), Z1 (one-layer zein fiber with ZMEO), Z3 (three-layer zein fiber with ZMEO), and Z5 (five-layer zein fiber with ZMEO).

**Figure 3 foods-13-00700-f003:**
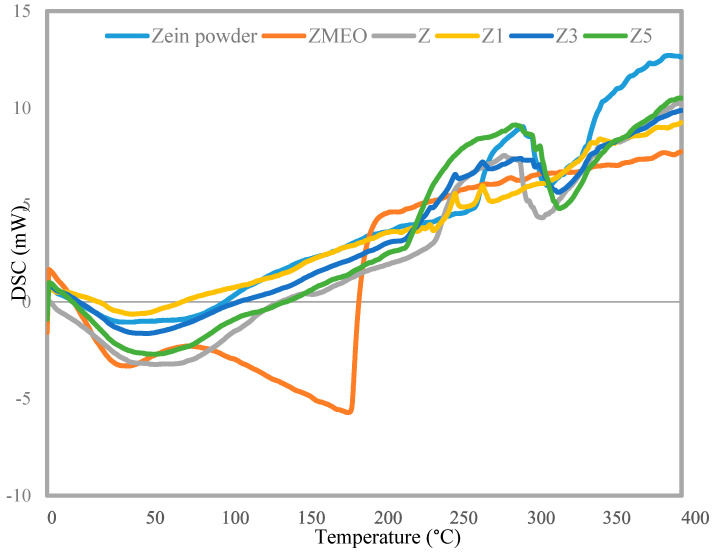
DSC thermograms for Z (zein fiber without ZMEO), Z1 (one-layer zein fiber with ZMEO), Z3 (three-layer zein fiber with ZMEO), and Z5 (five-layer zein fiber with ZMEO).

**Figure 4 foods-13-00700-f004:**
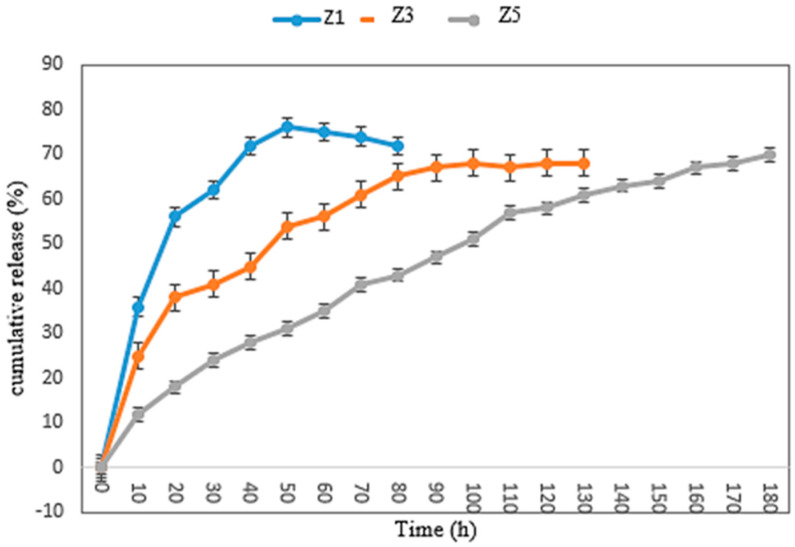
Profile of the release of ZMEO from Z1 (one-layer zein fiber with ZMEO), Z3 (three-layer zein fiber with ZMEO), and Z5 (five-layer zein fiber with ZMEO).

**Figure 5 foods-13-00700-f005:**
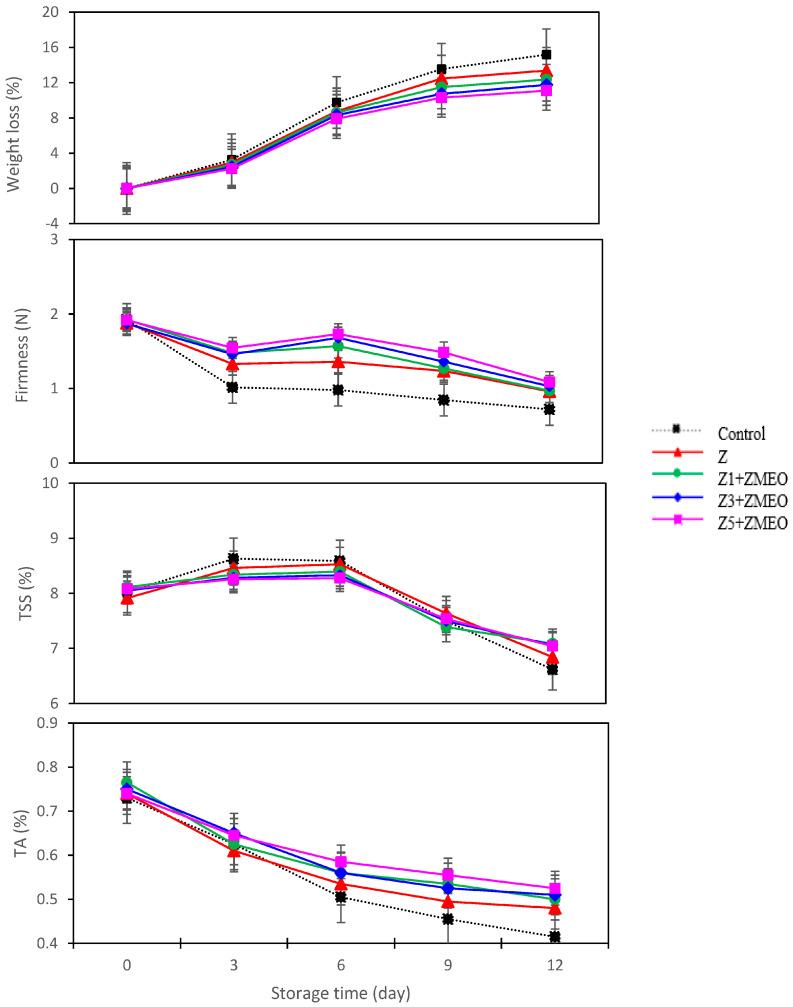
Interaction effect of active packaging using zein and zein/ZMEO fiber and storage time on weight loss (W.l), firmness, total soluble solids (TSSs), and titratable acidity (TA) in strawberries stored at 4 °C for 12 days.

**Figure 6 foods-13-00700-f006:**
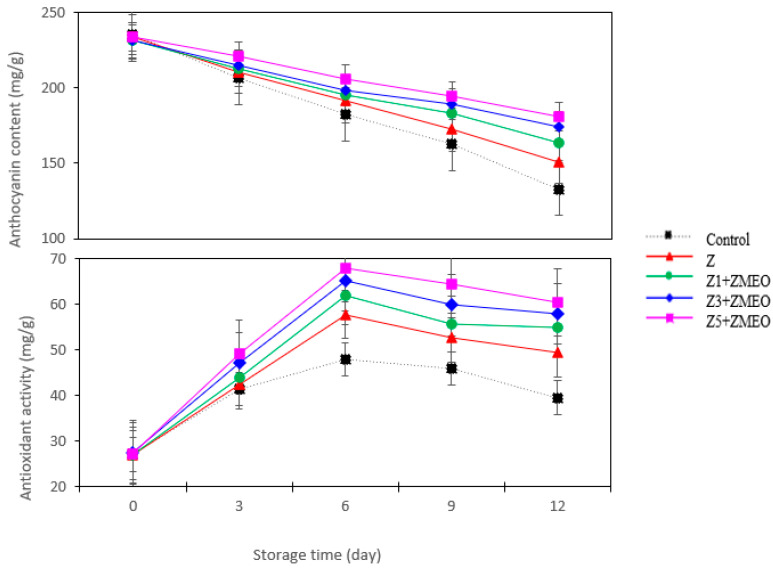
Interaction effect of active packaging using zein and zein/ZMEO fiber and storage time on anthocyanin content, and antioxidant activity in strawberries stored at 4 °C for 12 days. According to some researchers, after harvesting and storing strawberries, the process of anthocyanin biosynthesis remains active even at low temperatures [19]. Anthocyanins are a group of flavonoid compounds found in plants that contribute to the coloration of strawberry fruit [13]. Anthocyanin accumulation in higher plants, including active packaging that alters the packaging atmosphere, is a well-known response to environmental conditions [19]. In the present research, we found that the application of ZMEO enclosed with zein fibers in the packaging can increase the anthocyanin content in the post-harvest time of strawberry fruit. The results show that active packaging is a method to increase anthocyanin accumulation and maintain a high-quality product.

**Figure 7 foods-13-00700-f007:**
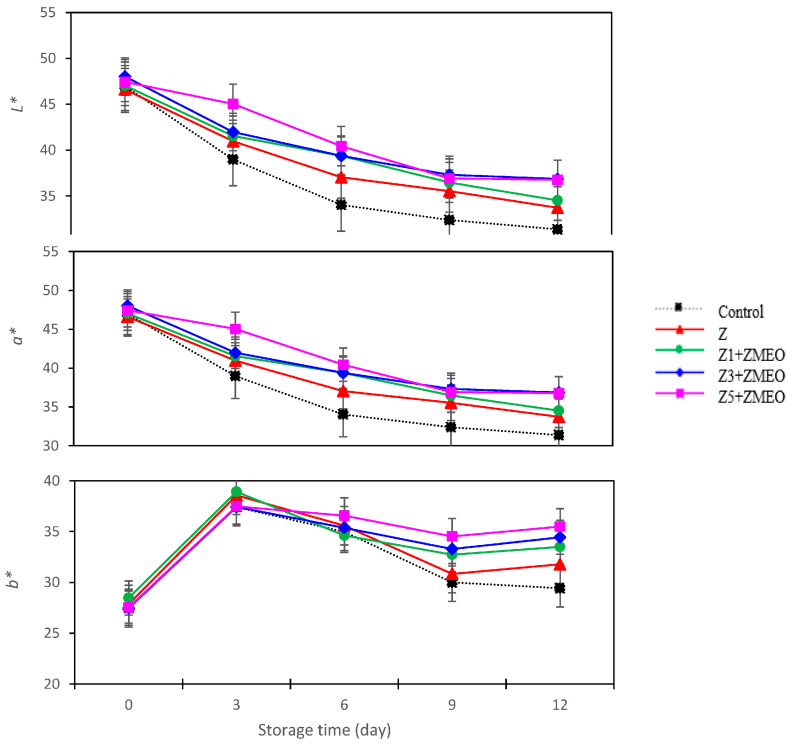
Interaction effect of active packaging using zein and zein/ZMEO fiber and storage time on color parameter (*L**, *a**, and *b**) in strawberries stored at 4 °C for 12 days.

**Table 1 foods-13-00700-t001:** Chemical and microbial characteristics and chemical components (%) identified through GC–MS.

Characteristic	Test Results	Acceptance Limit	Test Method
Appearance	Conform	Yellow to red color	Organoleptic
ColorOdor	Conform	Thyme odor
Chemical composition of essential oil	Concentration	RI	GC–MS analysis
α-Pinene	1.47	930
Camphene	0.03	944
β-Pinene	0.04	973
Myrcene	0.05	987
p-Cymene	14.32	1022
Limonene	0.24	1025
β-Ocimene	0.09	1043
Terpinene	22.46	1055
Saninebe hydrate	0.05	1063
Linalool oxide	0.1	1068
Terpinolene	0.02	1084
Linalool	4.96	1100
4-Terpineol	1.39	1174
Thymol	43.84	1290
Carvacrol	10.94	1300
Microbial specification			
Total aerobic microbial count (cfu/mL)	˂10	˂100	USP41
Total fungi and yeast count (cfu/mL)	˂10	˂10
*Pseudomonas aeruginosa*	Negative	Negative
*Staphylococcus aureus*	Negative	Negative
*Salmonella* sp.	Negative	Negative
*Escherichia coli*	Negative	Negative
*Candida albicans*	Negative	Negative

**Table 2 foods-13-00700-t002:** The EE and kinetic model parameters for ZMEO release in zein multilayer fiber.

Zein Multilayer	EE (%)	Model	R^2^	RMSE	Release Kinetic Model’s Data
Z1	43	Higuchi	0.796	2.813	k = 0.151
		Korsmeyer-Peppas	0.963	0.186	k = 0.197 n = 0.429
Z3	56	Higuchi	0.897	2.412	k = 0.099
		Korsmeyer-Peppas	0.966	0.234	k = 0.145 n = 0.406
Z5	82	Higuchi	0.981	2.121	k = 0.067
		Korsmeyer-Peppas	0.963	0.321	k = 0.026 n = 0.693

**Table 3 foods-13-00700-t003:** Effect of active packaging using zein and Z/ZMEO fiber on weight loss, firmness, total soluble solids (TSS), and titratable acidity (TA) in strawberries stored at 4 °C for 12 days.

Active Packaging Treatment (AP)	Weight Loss (%)	Firmness (N)	TSS (%)	TA (%)
Control	8.34 ± 0.74 ^a^	1.09 ± 0.01 ^d^	7.87 ± 0.72 ^a^	0.54 ± 0.01 ^d^
Z	7.50 ± 0.25 ^b^	1.35 ± 0.12 ^c^	7.87 ± 0.62 ^a^	0.57 ± 0.01 ^c^
Z1 + ZMEO	7.02 ± 0.22 ^c^	1.44 ± 0.14 ^b^	7.86 ± 0.71 ^a^	0.59 ± 0.02 ^b^
Z3 + ZMEO	6.64 ± 0.57 ^d^	1.48 ± 0.15 ^b^	7.83 ± 0.70 ^a^	0.59 ± 0.01 ^b^
Z5 + ZMEO	6.30 ± 0.54 ^e^	1.55 ± 0.12 ^a^	7.84 ± 0.73 ^a^	0.61 ± 0.00 ^a^
Significance	**	**	ns	**
LSD (0.05)	0.063	0.04	0.81	0.20
Storage time (T) (day)				
0	-	1.90 ± 0.10 ^a^	8.03 ± 0.78 ^b^	0.74 ± 0.02 ^a^
3	2.71 ± 0.02 ^d^	1.36 ± 0.11 ^c^	8.39 ± 0.54 ^a^	0.63 ± 0.05 ^b^
6	8.66 ± 0.58 ^c^	1.46 ± 0.11 ^b^	8.42 ± 0.85 ^a^	0.54 ± 0.04 ^c^
9	11.7 ± 0.84 ^b^	1.23 ± 0.08 ^d^	7.50 ± 0.52 ^c^	0.51 ± 0.02 ^d^
12	12.7 ± 0.72 ^a^	0.95 ± 0.10 ^e^	6.92 ± 0.24 ^d^	0.48 ± 0.01 ^e^
Significance	**	**	**	**
LSD (0.05)	0.021	0.08	0.82	0.41
Interaction AP × T	**	**	**	**

Data shown are the mean ± standard error of three replicates. Means within columns with the same letters are not significantly different. ns, non-significant; **, significant.

**Table 4 foods-13-00700-t004:** Effect of active packaging using zein and zein/ZMEO fiber on anthocyanin content, antioxidant activity, and color parameter (*L**, *a***,* and *b**) in strawberries stored at 4 °C for 12 days.

Active Packaging Treatment (AP)	Anthocyanin (mg 100 gFW)	Antioxidant Activity (mg/100 g)	Color Parameter		
			*L**	*a**	*b**
Control	138.99 ± 14.52 ^e^	40.40 ± 2.25 ^d^	36.75 ± 4.12 ^d^	31.85 ± 3.12 ^c^	19.36 ± 1.10 ^d^
Z	191.46 ± 12.41 ^d^	45.89 ± 2.54 ^c^	38.77 ± 2.54 ^c^	32.92 ± 3.22 ^bc^	20.38 ± 1.98 ^c^
Z1 + ZMEO	197.04 ± 16.65 ^c^	48.69 ± 3.12 ^b^	39.79 ± 3.84 ^b^	33.64 ± 3.00 ^ab^	20.67 ± 2.02 ^c^
Z3 + ZMEO	201.66 ± 16.57 ^b^	51.55 ± 4.24 ^b^	40.71 ± 4.23 ^a^	33.59 ± 3.25 ^ab^	21.38 ± 2.32 ^b^
Z5 + ZMEO	207.22 ± 14.20 ^a^	53.90 ± 4.98 ^a^	41.32 ± 3.73 ^a^	34.32 ± 3.45 ^a^	22.07 ± 2.00 ^a^
Significance	**	**	**	**	**
LSD (0.05)	1.39	0.84	1.45	1.13	0.68
Storage time (T) (day)					
0	233.21 ± 19.44 ^a^	27.16 ± 1.29 ^e^	47.21 ± 2.78 ^a^	27.76 ± 2.31 ^d^	26.61 ± 2.15 ^a^
3	213.12 ± 24.09 ^b^	44.86 ± 3.89 ^d^	41.70 ± 3.58 ^b^	37.96 ± 3.24 ^a^	23.31 ± 2.651 ^b^
6	194.52 ± 20.54 ^c^	60.18 ± 4.22 ^a^	38.05 ± 3.81 ^c^	35.42 ± 2.87 ^b^	20.92 ± 2.42 ^c^
9	180.32 ± 18.68 ^d^	55.76 ± 4.07 ^b^	35.72 ± 3.30 ^d^	32.27 ± 3.01 ^c^	17.05 ± 1.87 ^d^
12	160.49 ± 17.45 ^e^	52.47 ± 5.55 ^c^	34.65 ± 3.22 ^e^	32.92 ± 3.24 ^c^	15.97 ± 1.88 ^e^
Significance	**	**	**	**	**
LSD (0.05)	1.35	1.85	1.24	2.52	0.75
Interaction AP × T	**	**	**	**	**

Data shown are the mean ± standard error of three replicates. Means within columns with same letters are not significantly different. ns: non-significant, **: significant.

## Data Availability

Data are contained within the article.

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
