# Peer review of "Zein Multilayer Electrospun Nanofibers Contain Essential Oil: Release Kinetic, Functional Effectiveness, and Application to Fruit Preservation"

_foods, 2024, doi:10.3390/foods13050700_

Round 1

Reviewer 1 Report

Comments and Suggestions for Authors

The manuscript entitled “Zein multilayer electrospun nanofibers contain essential oil: release kinetic, functional effectiveness and application to fruit preservation” by Moradinezhad et al. is an interesting paper since it is important to search for natural film coatings as alternatives, to extend the shelf life of fruits.

The authors evaluate numerous parameters to characterize the nanofibers and then the performance of Zein/ZMEO nanofiber film in prolonging the shelf life and maintaining the quality of strawberries, which makes this work more robust and present appropriate research design to develop the work. The paper is well discussed, with relevant literature review.

The authors should pay attention to the following comments:

GENERAL

The paper should present the line numbers, because it is easier to report the errors.

Please check the manuscript to confirm if the structure of the paper is according to the journal guidelines. Some points are not correct (e.g. in the text, figure instead of “fig”, or the references). Furthermore, the authors should revise grammatically all the manuscript.

When analyzing the effect of active packaging in different parameters, why the value of the control and the value of time 0 days in storage time is not the same?

INTRODUCTION

“the strawberry (Fragaria ananassa)”: the “the” has to be “The” and “Fragaria ananassa” has to be in italic, please.

“Essential oils have antibacterial, antifungal, and antioxidant properties. The Food and Drug Administration (FDA) considers them safe and they can be used as natural food addi-tives and can prevent the growth of harmful bacteria.”: I would like to see here a reference.

“is native to Iran”: “is native from Iran”.

M&M

Evaluation of strawberry quality

“The digital balance from (UWA-K-015, China)”: please reformulate

“total phenols”: change to total phenolic content.

“The results were reported as milligrams per liter of Gallic acid equivalents (GAE) per 100 grams of fresh weight”: please explain this.

R&D

FTIR

Line 2-9: please reformulate grammatically these sentences.

DSC

Line 1-2: reformulate to “The DSC thermograms of powder zein, ZMEO and multilayer nanofiber (Z, Z1, Z3 and Z5) are demonstrated in Figure 3.”

Encapsulation efficiency (EE) and Release Analysis and Kinetic Modeling

Figure 4: the caption on the figure says z1, z2 and z3 and not z1, z3 and z5.

Line 12: “Zhang et al (2020) reported that the control group showed rapid evaporation”

The effect of active packaging on weight loss percentage

“The characteristics of the fruit skin have a significant effect on the amount of weight loss. Because, the process of losing weight in fruits is primarily linked to the respiration and moisture evaporation that occurs through their skin (Shehata et al., 2020).” Please reformulate.

The effect of active packaging on firmness (N)

Is the storage period 12 days or 15 days as it was reported in the M&M?

“Castillo et al. (2012) showed that packaging with carvacrol, eugenol, and thymol essential oils maintained the firmness of plum fruit tissue (about 20% more than the control), they explained that the essential oils used probably affect the ethylene intermediate compounds. and in this way, they delay the ripening of the fruit, which preserves the firmness of the texture.” Please reformulate.

The effect of active packaging on total soluble solids (TSS)

“As shown in Fig. 5, examining the interaction effects of storage time and active packaging indicated that although the amount of TSS decreased during storage.” Please reformulate to be clarified.

“The results of the present research or the results of Raafat et al. (2012) and Geransayeh et al. (2015) are consistent.”: change to “The results of the present research are consistent with the results of Raafat et al. (2012) and Geransayeh et al. (2015).”

The effect of active packaging on titratable acidity (TA)

“The numerical range of strawberry TA in this research was reported as 0.4 to 0.8%.” where are these results?

The effect of active packaging on the antioxidant activity

In table 4, the authors should mention “antioxidant activity” instead of “antioxidants”. The same in the text. You are investigating the antioxidant activity and not the antioxidant content.

“The debatable scenario in this case is that the release of thymol from the zein fibers during storage reduces the contamination and deterioration of the fruit, so the respiration of the fruit decreases and finally the amount of oxygen consumption, which is an important factor in the oxidative (Kazemi-Pasarvi et al., 2020).” Please reformulate.

The effect of active packaging on colour parameter (L*, a* and b*)

“Probably, in the current research, thymol enclosed in zein layers reduces free radicals and prevents the destruction of anthocyanin and reducing the level of a* in the paint.” – by the results in the table, the values of a did not decrease. Are you talking about figure 7?

Conclusion

The conclusion is very general. The authors should report more specific results.

Author Response

Many thanks for your useful comments and suggestions about our manuscript.

We have modified the manuscript accordingly; all reviewers’ comments are answered accurately. Some parts of the manuscript have been changed and some sections have been explained more. Detailed corrections are listed below point by point:

Comments

Operations

Reviewer 1

GENERAL

The paper should present the line numbers, because it is easier to report the errors.

Done

Please check the manuscript to confirm if the structure of the paper is according to the journal guidelines. Some points are not correct (e.g. in the text, figure instead of “fig”, or the references). Furthermore, the authors should revise grammatically all the manuscript.

Done

When analyzing the effect of active packaging in different parameters, why the value of the control and the value of time 0 days in storage time is not the same?

The value of time = the average of the data of all treatments on day zero

The value of control on zero day= the average of the control sample data on day zero

But there was no significant difference between the average data of all treatments on day zero and the control on day zero.

INTRODUCTION

“the strawberry (Fragaria ananassa)”: the “the” has to be “The” and “Fragaria ananassa” has to be in italic, please.

Done

“Essential oils have antibacterial, antifungal, and antioxidant properties. The Food and Drug Administration (FDA) considers them safe and they can be used as natural food addi-tives and can prevent the growth of harmful bacteria.”: I would like to see here a reference.

Was added

“is native to Iran”: “is native from Iran”.

Was corrected

M&M

Evaluation of strawberry quality

Done

“The digital balance from (UWA-K-015, China)”: please reformulate

Done

R&D

FTIR

Line 2-9: please reformulate grammatically these sentences.

Was written

DSC

Line 1-2: reformulate to “The DSC thermograms of powder zein, ZMEO and multilayer nanofiber (Z, Z1, Z3 and Z5) are demonstrated in Figure 3.”

Was corrected

Encapsulation efficiency (EE) and Release Analysis and Kinetic Modeling

Figure 4: the caption on the figure says z1, z2 and z3 and not z1, z3 and z5.

Was rewritten

Line 12: “Zhang et al (2020) reported that the control group showed rapid evaporation”

was corrected

The effect of active packaging on weight loss percentage

“The characteristics of the fruit skin have a significant effect on the amount of weight loss. Because, the process of losing weight in fruits is primarily linked to the respiration and moisture evaporation that occurs through their skin (Shehata et al., 2020).” Please reformulate.

Was rewritten

The effect of active packaging on firmness (N)

Is the storage period 12 days or 15 days as it was reported in the M&M?

The physicochemical properties of the fruits were measured at different intervals (day 0, 3, 6, 9, and 12).

“Castillo et al. (2012) showed that packaging with carvacrol, eugenol, and thymol essential oils maintained the firmness of plum fruit tissue (about 20% more than the control), they explained that the essential oils used probably affect the ethylene intermediate compounds. and in this way, they delay the ripening of the fruit, which preserves the firmness of the texture.” Please reformulate.

Was rewritten

The effect of active packaging on total soluble solids (TSS)

“As shown in Fig. 5, examining the interaction effects of storage time and active packaging indicated that although the amount of TSS decreased during storage.” Please reformulate to be clarified.

Was rewritten

“The results of the present research or the results of Raafat et al. (2012) and Geransayeh et al. (2015) are consistent.”: change to “The results of the present research are consistent with the results of Raafat et al. (2012) and Geransayeh et al. (2015).”

Was rewritten

The effect of active packaging on titratable acidity (TA)

“The numerical range of strawberry TA in this research was reported as 0.4 to 0.8%.” where are these results?

The numerical range of strawberry TA in this research was reported as 0.54 to 0.61%.

The effect of active packaging on the antioxidant activity

In table 4, the authors should mention “antioxidant activity” instead of “antioxidants”. The same in the text. You are investigating the antioxidant activity and not the antioxidant content.

The whole text was corrected

“The debatable scenario in this case is that the release of thymol from the zein fibers during storage reduces the contamination and deterioration of the fruit, so the respiration of the fruit decreases and finally the amount of oxygen consumption, which is an important factor in the oxidative (Kazemi-Pasarvi et al., 2020).” Please reformulate.

Was rewritten

The effect of active packaging on colour parameter (L*, a* and b*)

“Probably, in the current research, thymol enclosed in zein layers reduces free radicals and prevents the destruction of anthocyanin and reducing the level of a* in the paint.” – by the results in the table, the values of a did not decrease. Are you talking about figure 7?

Following this reaction, the levels of a* (red) decrease drastically. Probably, in this research, ZMEO enclosed in zein layers reduces free radicals and prevents the destruction of anthocyanin and reducing the level of a* of strawberries. Our findings were consistent with the results of other researchers

Conclusion

The conclusion is very general. The authors should report more specific results.

The conclusion was rewritten and a further extension of the results was given

Reviewer 2

Abstract

1.     There is a potential discrepancy in the reported time frames: the study reports on a 12-day test period of antibacterial properties but then later mentions a 15-day period of cold storage.

Was corrected

2.     4â—¦C should be 4ºC (check whole manuscript)

The whole text was corrected

Manuscript

1.          Number each subparagraph e.g. Solution preparation, Electrospining

Was added

2.     Add name of paragraph Chromatographic method

Was added

3.     More information about chromatography is needed:

a.     What was the injection mode? Direct? SPME? If direct which volume?

b.     Temperature of injector

c.     Oven program (temperatures)

d.     Splitless or split mode?

e.     m/z range

f.      type of library used for identification of compounds.

The GC/MS analyses were executed on a Varian CP-3400 gas 119 chromatograph equipped with a column BPX5 (30 m × 0.25 mm × 0.25um, 120 SGE (Scientific Instrument Service, NJ, USA) coupled to Saturn 2000 mass 121 spectrometer. The column temperature was programmed at 50 ºC as an 122 initial temperature, holding for 2 min, with 2.5 ºC increases per minute to 123 265 °C. Injection port temperature was 250 ºC and helium was used as 124 carrier gas at a flow rate of 1 mL/min. Ionization voltage of mass 125 spectrometer in the electron impact mode was equal to 70 eV and ionization 126 trap temperature was 170 ºC. The mass spectrometer was scanned from m/z 127 40 to 250. The individual compounds were identified and confirmed 128 there after using Kovats retention indices. Pure standards of the compounds 129 were also used to confirm the identification of the compounds.

4.     The entrapment efficiency (EE) of the ZMEO–loaded nanofiber was indirectly deter- mined to dissolve the film (5 gr), should be 5 g

Was corrected

 5.     The strawberries were treated by immersing them in a 0.1% sodium hypochlorite solution for 1 min, followed by dripping off for 5 min. the strawberries were… second phrase should start with capital letter

Was corrected

6.     In paragraph Fruit preparation and active packaging add photo of packaging wih strawberry

There were no suitable pictures to be included in the scientific article, only a few pictures were sent for clarification

7.     Evaluation of strawberry quality: write each method separately Total phenols, DPPH…

Was separated

8.     Table 1 add new table with retention time, CAS and chemical structure for obtained compounds by GC-MS

Was added

9.     Describe in a method paragraph how did you do quantitative analysis, which standards did you used.

The distinction between samples was assessed using analysis of variance (One-way ANOVA with Duncan test) at a significance level of p < 0.05. A post hoc test for independent samples was conducted using the statistical software SPSS (IBM SPSS Statistics, Version 22, New York, USA).

10.  Table 1 in concentrations you need to put name ´Concentration´and unit.

Was corrected

11.  Figures 5, 6, 7 add error bars.

Error baes were added to all figures

12.  Figure caption of figure 7 is cut.

Was corrected

The manuscript has been resubmitted to your journal. We look forward to your positive response.

Reviewer 2 Report

Comments and Suggestions for Authors

The study presents a commendable innovation in the field of food preservation through the use of advanced material science and encapsulation techniques. The employment of sequential electrospinning to create a multilayer film incorporating zein nanofibers and Zataria multiflora essential oil is a sophisticated approach to enhance the shelf life of perishable food items, such as strawberries. The meticulous structuring of the nanofibers to achieve different layering with variable release profiles is particularly very interesting and shows a high level of control over the material properties.

The observation from scanning electron microscopy that the fibers were uniform and smooth is indicative of a successful electrospinning process, ensuring quality and consistency in the films produced. The seamless integration of ZMEO within the multilayer Z without any beads or irregularities is indicative of the careful optimization of the electrospinning parameters, which is crucial for the creation of effective barrier properties. The article is well written, but I have some comments to improve it.

Abstract

1.     There is a potential discrepancy in the reported time frames: the study reports on a 12-day test period of antibacterial properties but then later mentions a 15-day period of cold storage.

2.     4â—¦C should be 4ºC (check whole manuscript)

Manuscript

1.      Number each subparagraph e.g. Solution preparation, Electrospining

2.     Add name of paragraph Chromatographic method

3.     More information about chromatography is needed:

a.     What was the injection mode? Direct? SPME? If direct which volume?

b.     Temperature of injector

c.     Oven program (temperatures)

d.     Splitless or split mode?

e.     m/z range

f.      type of library used for identification of compounds.

4.     The entrapment efficiency (EE) of the ZMEO–loaded nanofiber was indirectly deter- mined to dissolve the film (5 gr), should be 5 g

5.     The strawberries were treated by immersing them in a 0.1% sodium hypochlorite solution for 1 min, followed by dripping off for 5 min. the strawberries were… second phrase should start with capital letter

6.     In paragraph Fruit preparation and active packaging add photo of packaging wih strawberry

7.     Evaluation of strawberry quality: write each method separately Total phenols, DPPH…

8.     Table 1 add new table with retention time, CAS and chemical structure for obtained compounds by GC-MS

9.     Describe in a method paragraph how did you do quantitative analysis, which standards did you used.

10.  Add analytical parameters of GC method such as linearity and LOD and LOQ of each compound.

11.  Table 1 in concentrations you need to put name ´Concentration´and unit.

12.  Figures 5, 6, 7 add error bars.

13.  Figure caption of figure 7 is cut.

Author Response

(The authors gave the same response as above.)

Round 2

Reviewer 2 Report

Comments and Suggestions for Authors

All my comments have been addressed. The manuscript has been improved significantly.